# Factors Associated with Controlled Glycemic Levels in Type 2 Diabetes Patients: Study from a Large Medical Center and Its Satellite Clinics in Southeast Region in the USA

**DOI:** 10.3390/healthcare12010026

**Published:** 2023-12-21

**Authors:** Tran Ha Nguyen, Gianluca De Leo, Amanda Barefield

**Affiliations:** 1Department of Health Management, Economics and Policy, Augusta University, Augusta, GA 30912, USA; gdeleo@augusta.edu; 2Department of Undergraduate Health Professions, Augusta University, Augusta, GA 30912, USA; abarefield@augusta.edu

**Keywords:** diabetic care, glycemic control, type 2 diabetes, comorbidities, chronic disease

## Abstract

Diabetes, including type 1, type 2, and gestational, is a significant public health issue responsible for various clinical, economic, and societal issues. Most of the consequences, if uncontrolled, can result in serious health problems, such as heart disease, vision loss, and kidney disease. Approximately 37.3 million Americans have diabetes, including 37.1 million adults 18 years or older, with 90–95% type 2 diabetes (T2D). The purpose of this study is (1) to explore the profile of patients with T2D and (2) to identify the associated factors of diabetic status. Examined factors included sociodemographic characteristics, social factors, and comorbidities. The study analyzed a primary dataset from a retrospective chart review of adult patients with T2D who were seen at a large medical center and its satellite clinics in the southeast region of the United States in 2019. Sex, dyslipidemia, and the number of concordant comorbidities were found to be significant associated factors of diabetic status. In the era of intertwined patient-centered approach and public health, the study’s findings can guide treatment plans and interventions targeting individuals and communities.

## 1. Introduction

Diabetes mellitus (DM), commonly known as diabetes, is a significant public health issue responsible for various clinical, economic, and societal issues [1,2]. Most of the consequences of diabetes, if uncontrolled, can result in serious health problems, such as heart disease, vision loss, and kidney disease [3]. Since 2000, diabetes has been among the top ten leading causes of death worldwide [4]. Diabetes is also a significant risk factor for developing cardiovascular disease and stroke, the leading causes of death since the early 1900s [5]; in addition, diabetes represents a substantial burden to healthcare systems. The International Diabetes Federation (IDF) estimated that 537 million people worldwide had diabetes in 2021, resulting in health expenditures of 966 billion American dollars globally. The IDF forecasted that the financial burden of diabetes will reach more than 1054 billion dollars by 2045 [5,6]. In the United States (U.S.), approximately 37.3 million Americans, or 11.3% of the U.S. population, have diabetes, including 37.1 million adults 18 years or older, or 14.7% of all U.S. adults [7]. Each year, the U.S. spends 237 billion dollars on direct medical costs related to diabetes and another 90 billion dollars on reduced work productivity [8].

Diabetes is characterized by high blood glucose levels resulting from the effects of abnormal beta cell biology on insulin action [3]. Diabetes is classified into type 1 diabetes, type 2 diabetes (T2D), and gestational diabetes [9]. About 90–95% of all diabetes are T2D, with most individuals having at least one comorbidity that can affect their self-care abilities [10,11]. Even though diabetes can lead to other serious health problems, there are means to prevent or delay these complications, such as medication and/or lifestyle/behavior changes, especially in T2D patients [7]. The literature provides evidence that controlling blood glucose within the optimal range would likely reduce the risk of complications in T2D patients [12,13,14]. Blood glucose is often assessed by the Hemoglobin A1C (A1C) measurement, which reflects average glycemia over approximately three months. The American Diabetes Association (ADA) recommends the goal for A1C measurement for nonpregnant diabetic adult patients to be less than 7% (or 53 mmol/mol) without significant hypoglycemia [15,16].

Disparities in diabetes are well documented. Previous studies have demonstrated that diabetes affects racial and ethnic minorities and low-income adult populations in the U.S. disproportionately, with relatively intractable patterns seen in these populations’ higher risk of diabetes complications and mortality [17]. About 35 reported T2D-related comorbidities include hypertension, cancer, kidney disease, cardiovascular diseases, and overweight and obesity [18,19]. Diabetes-concordant comorbidities are typically referred to as cardiovascular and metabolic conditions, such as dyslipidemia because they share parts of the same overall pathophysiologic profile as T2D [10]. Dyslipidemia is the condition of an imbalance of lipid levels with elevated blood levels, such as cholesterol, low-density lipoprotein (LDL), triglycerides, and high-density lipoprotein (HDL) [20]. Diabetes-discordant comorbidities do not have any direct relationship with T2D and include chronic conditions such as asthma, cancer, and mental illness [21]. Social factors, such as drinking, smoking, and substance abuse, have been studied in relation to T2D. Cigarette smoking has been linked to an increased risk of T2D [22]. A decrease in the risk of T2D among moderate alcohol drinkers may be confined to women and non-Asian populations [23]. A literature review provided evidence that early-life exposure to substance abuse may be responsible for increasing the risk of T2D [24]. 

On the population level, several research studies have focused on the prevention and delay of diabetes among individuals with prediabetes rather than the prevention and delay of complications among patients with diabetes. National data presented the challenge of patients with diabetes to achieve the target A1C level (<7%) in controlling their condition [25]. While most patients with diabetes have at least one comorbidity, meeting the glycemic target is crucial, as chronically uncontrolled diabetes is associated with increased complications from comorbidities [12,13,14]. The purpose of this study is (1) to explore the profile of patients with T2D and (2) to identify the associated factors of controlled glycemic levels in T2D, including sociodemographic characteristics, social factors, and comorbidities. The study analyzed a primary dataset from a retrospective chart review of adult patients with T2D who were seen at a large health system, Augusta University Health System (AUH), and its satellite clinics in the southeast region of the U.S. in 2019. Patients visiting AUH are from across the State of Georgia and parts of the State of South Carolina. Georgia ranks 32nd in the nation for diabetes according to America’s Health Rankings, while South Carolina ranks 39th [26]. Hereafter, in this manuscript, the words “*diabetes*” and “*T2D*” are used interchangeably to indicate *type 2 diabetes*. 

## 2. Methods

### 2.1. Data Source

The study used a quantitative, cross-sectional observational design based on primary data. Data were from retrospective patient record reviews of adults with T2D who visited the Augusta University Health System’s clinics and satellites in 2019 for various reasons. Situated in Augusta, Georgia’s second oldest and second-largest city, Augusta University Health System (AUH) is affiliated with Augusta University (AU). It offers the most advanced healthcare services via a 478-bed adult hospital, the 154-bed Children’s Hospital of Georgia, the Georgia Cancer Center, and more than 80 outpatient clinics across Georgia and South Carolina. 

### 2.2. Study Population and Sample

In 2019, 4071 adults (aged 18 or above) diagnosed with T2D visited AUH for various reasons, such as primary/follow-up care, specialty care, pre/post-surgery, and so on. The demographic characteristics were as follows: 1960 (48.1%) male and 2111 (51.9%) female; 57 (1.4%) in the 18–24 years age group, 153 (3.8%) in the 25–34 age group, 351 (8.6%) in the 35–44 age group, 745 (18.3%) in the 45–54 age group, 1054 (25.9%) in the 55–64 age group, and 1711 (42.0%) in the 65 years and above age group; 1894 (46.5%) as White, 1962 (48.2%) as Black, 82 (2.0%) as Hispanic, 51 (1.3%) as Asians, and 82 (2.0%) as other races or multiracial. 

An estimated minimum sample size of 193 was computed using online OpenEpi version 3.01 [27]. The study used the module “Sample Size for a Proportion or Descriptive Study” with the following parameters: population size = 1 million (as large to unlimited) to ensure a good sample size for data analysis, anticipate % frequency = 14.7% (as the adult American diabetes prevalence), confidence limit = 5% (confidence level at 95%), and design effect for complex sample survey = 1 (random sample). Microsoft Excel 2021 [28] systematic random sampling tool was utilized to select 206 patients for chart reviews. 

### 2.3. Measurements

Patient information obtained by performing record reviews included hemoglobin A1C level, demographic characteristics (age, sex, race, insurance type, metro/nonmetro), and comorbidities (body mass index (BMI), hypertension, cardiovascular disease, dyslipidemia presenting abnormally elevated levels of blood lipids and cholesterol, kidney disease, and cancer). Self-reported patient information related to smoking and alcohol habits and substance abuse were also collected. 

Utilizing the patient’s A1C levels, the status of T2D was coded as controlled diabetes when the value of A1C levels was less than 7% and uncontrolled diabetes when the value was equal to or greater than 7% [15,16]. The A1C level was measured using ion-exchange high-performance liquid chromatography (HPLC) on the Variant II Turbo Hemoglobin Testing System, and the analytical range (NGSP) is from 3.4 to 20.6% [29].

The age variable was categorized in the following groups: 18–24, 25–34, 35–44, 45–54, 55–64, and 65 years and above. The sex variable included males and females. The race variable was categorized as White, Black, Hispanic, Asian, or other, including multiracial. The insurance type variable was categorized into the following groups: private, Medicare, government-assisted plan, and no insurance. The metro/nonmetro variable was derived from the patient’s residency zip code, which determined the patient’s county of residence. The study used the Rural-Urban Commuting Area Codes (RUCA) to code the county to metro or non-metro [30]. Obesity was determined by recoding the patient’s BMI as normal weight (BMI ≤ 25), overweight (25 > BMI ≤ 29), and obese (BMI > 29) [31]. The status of other comorbidities and social habits, including smoking, alcohol use, and substance abuse, were recorded from the record review (no or yes). Obesity, hypertension, cardiovascular disease, and dyslipidemia were examined as T2D concordant comorbidities, while kidney disease and cancer were T2D discordant comorbidities. The concordant comorbidity variable was the sum of four concordant comorbidities, while the discordant and total comorbidity variables were the sum of two discordant and six comorbidities, respectively. 

The study’s dependent variable is diabetes status (uncontrolled versus controlled), while demographic characteristics, comorbidities (individual, concordant, discordant, and total comorbidity), and social habits are independent variables. 

### 2.4. Data Analysis

Data analysis was performed using IBM SPSS version 27.0 [32]. Descriptive statistics and crosstabulations were performed on variables as appropriate. Multivariable logistic regression was utilized to determine the association between diabetes status and demographic characteristics, comorbidities, and social habits. The threshold of statistical significance was set at *p* < 0.05. 

## 3. Results

Table 1 details the demographics, comorbidities, and social habits of T2D patients. Of 206 patients with T2D in the study sample, 110 (53.4%) controlled their diabetes, while 96 (46.6%) did not. The sex of patients with T2D was almost equally distributed between males (51.0%) and females (49.0%), with slightly more male predominance; however, more female (62.4%) than male (44.8%) patients had controlled diabetes. The age group “over 65 years old” had the most significant number of T2D patients (40.8%). The proportions of controlled versus uncontrolled T2D showed no significant difference among the age groups up to 64 years. Nearly half of the patients (46.4%) with controlled diabetes were in the age group “over 65 years old”, while 34.4% of patients with uncontrolled diabetes were in the same age group. 

White and Black patients accounted for more than 97% of all diabetes, while 3% presented all other races (Hispanics, Asians, and others). More than half of patients with controlled diabetes were White (51.8%), followed by Black (45.5%), and other races (2.7%). The majority of patients with T2D had either private insurance or Medicare. While Medicare patients had the highest proportion of controlled diabetes (39.1%), patients with private insurance had the highest proportion of uncontrolled diabetes (41.7%). Most individuals with T2D (92.7%) live in the metro area, accounting for 95.5% of the controlled T2D and 89.6% of the uncontrolled T2D. 

Among T2D-associated comorbidities, hypertension was found to be the most prevalent among patients with T2D, accounting for 82.5% among all patients, 80.9% in controlled patients with T2D, and 84.4% in patients with uncontrolled T2D, even though there was no significant difference between the two groups of patients. Over half of the patients with T2D were obese (59.7% in all patients with T2D, 55.5% in patients with controlled T2D, and 64.6% in patients with uncontrolled T2D), and approximately one-quarter were overweight (24.3% in all patients with T2D, 28.2% in patients with controlled T2D, and 19.8% in patients with uncontrolled T2D). Other comorbidities (cardiovascular disease, dyslipidemia, kidney disease, and cancer) and social habits had a similar proportion among patients with both controlled and uncontrolled T2D. 

While approximately 5% of patients did not have another chronic condition (4.9% all, 5.2% controlled, and 4.5% uncontrolled), approximately 6% did not have any concordant comorbidity (6.8% all, 8.2% controlled, and 5.5% uncontrolled), and over 66% did not have any discordant comorbidity (67.58% all; 66.4% controlled, and 68.8% uncontrolled). 

Table 2 presents the results of the adjusted logistic regression model for the associations between diabetic status (uncontrolled vs. controlled) and sociodemographics, social habits, and comorbidities in T2D patients. Sex was significantly associated with diabetic status, with females being 2.673 times more likely to have controlled T2D than their counterparts. Dyslipidemia was significantly associated with diabetic status. Patients with T2D who also had dyslipidemia were 0.495 times less likely to have controlled diabetes compared to those with normal lipidemia. 

Figure 1 illustrates the logistic model for the relationship between diabetic status and sociodemographics, social habits, and comorbidities, demonstrating the statistical significance of the associated factors of controlled diabetes.

Table 3 demonstrates the relationship of diabetic status with concordant and discordant comorbidities. Concordant comorbidities were also found to be associated with T2D status. The more concordant comorbidities T2D patients had, the lower the odds they had controlled diabetes, with a noted 0.741 times for each additional diabetic concordant condition/disease. 

## 4. Discussion

In this study, primary data from a large health system, serving residents of the Southeast region in the U.S., were used to explore the profile of patients with T2D and to identify the associated factors of diabetic status (controlled versus uncontrolled), including sociodemographic characteristics, social factors, and comorbidities. A typical controlled diabetic patient can be defined through our findings as “an older obese white female with hypertension who lives in the metro area”.

Sex is a significant associated factor of diabetic status, with females being more likely to have controlled diabetes. This finding is congruent with diabetic studies in the U.S. and Canada, which suggested that women are more proactive than men about managing their T2D [33,34]. Controlling diabetes is defined as utilizing health services to comply with diabetic treatment and self-monitoring blood glucose. Females are well-documented in the literature to be more likely to participate in health screenings and services for other chronic disease/condition management as well as in general [35,36]. Nevertheless, our finding conflicted with studies in Italy, Brazil, and Venezuela, which found that women with T2D had worse glycemic control than men [37,38]. There were suggestions for contributing factors related to the gender difference in glucose homeostasis and sociocultural in different populations. This information is crucial in caring for patients with T2D and designing diabetes prevention programs, accounting for the sex factor and their behaviors/attitudes toward self-managing their condition. 

Another associated factor of diabetic status noted in the study is dyslipidemia. Patients with T2D with co-existing dyslipidemia are found to be less likely to have controlled diabetes, lower than half of the odds (AOR = 0.495). The literature provides evidence that LDL, HDL, and triglycerides are independent predictors of cardiovascular disease, the number one killer worldwide since the early 1990s. Although the relationship between dyslipidemia and diabetes is not fully understood, evidence from the literature suggests that dyslipidemia is somehow related to poor glycemic control, insulin resistance, inflammation, and genetic susceptibility, and interventions to improve glycemia usually lower lipid levels [39]. Our finding reinforced the call for lipoprotein screening among T2D patients to treat the condition appropriately. Evidence-based studies have confirmed that controlling lipid levels is beneficial in patients with T2D, reducing the event rate for major CVD events [40]. 

The number of diabetic concordant comorbidities is also found to be significantly associated with controlled diabetic status. Each additional concordant comorbidity decreased the odds by 0.741 of having controlled diabetes. This phenomenon can be plausibly explained by the shared risk factors among diabetes and its concordant comorbidities. The underlying molecular mechanisms contribute to the close relationship between diabetes and its concordant comorbidities [41]. A previous research study that followed a cohort of T2D patients for 11 years from the onset of the disease reported that concordant comorbidities for T2D are considered expected outcomes of the disease or disease complications [42]. Another 16-year study that observed the development of T2D comorbidities of newly diagnosed T2D found that the effectiveness of the patient-centered intervention was not different among patients who developed comorbidities compared to those who did not [43]. The relationship between T2D and its concordant comorbidities warrants further research. 

Other independent variables in the logistic model were not statistically associated with T2D status, even though they are closely related to T2D, specifically hypertension. Hypertension accounts for 82.5% of our sample, 80.9% of controlled T2D, and 84.4% of uncontrolled T2D. There was a slight relative indifference between the two groups of the T2D status. Hypertension and T2D are closely associated; however, the causal effect between the two remains unclear. They are the two cardiovascular risk factors most prevalent in the global population. Future studies should disentangle the complex network of molecular pathways that link these two comorbidities. 

Health insurance type, including no insurance, was not statistically associated with glycemic control in our study. Health insurance is well-known to enable access to care and improve health equity [44]. However, the association between health insurance coverage and controlled glycemic levels is poorly documented. Nelson et al. (2005) analyzed the 2000 Behavioral Risk Factor Surveillance System to examine the association between the type of health insurance coverage and the quality of care provided to individuals with diabetes in the U.S. Even though the findings indicated an association, the factors examined were self-reported compliance behavior with diabetic care guidelines rather than measured blood levels of A1C [45]. Gold and colleagues (2021) found having health insurance of any type was associated with significantly lower fasting glucose for individuals with diabetes [46]. According to The American Diabetes Association (ADA), a blood glucose test may indicate hyperglycemia as uncontrolled diabetes, while an A1C test does not. The A1C measurement was recommenced as a primary test for diabetic diagnosis and confirming diabetic status [47]. Perhaps the impact of having insurance on glycemic control was reduced by the effects of other sociodemographic and behavioral factors and comorbidities. Future studies may shed light on this relationship. 

Using the primary dataset from a large hospital in the southeast region of the U.S. is a strength of our study. Other than the self-reported information on social factors (smoking, drinking, and substance use), clinical information was obtained from patient records, providing the accuracy of the findings and avoiding some recall bias. Nevertheless, the study contains limitations that should not be overlooked. First, a cross-sectional study design was used; thus, the study cannot establish the temporal relationship between T2D and its associated factors. No causation can be determined. Second, clinical information from patient records was retrieved, not specifying patients’ duration of diabetes or whether patients were under a treatment plan or intervention. The logistics regression model was not adjusted for diabetic duration, medication, and/or a healthy lifestyle program, which may positively influence patients’ A1C levels. Lastly, the study sample is from one medical center, limiting the findings’ generalization. 

## 5. Conclusions

The increasing prevalence of type 2 diabetes and the extensive and diverse types of morbidity have become an urgent health priority. In the era of intertwined patient-centered approach and public health, the study’s findings can potentially guide treatment plans and interventions targeting individuals and communities to improve diabetic outcomes. Information from the study can aid policymakers and healthcare providers/educators in their decision-making to bolster programs that benefit patients and the public’s well-being, as well as reduce T2D’s significant economic consequences on health systems and the nation.

## Figures and Tables

**Figure 1 healthcare-12-00026-f001:**
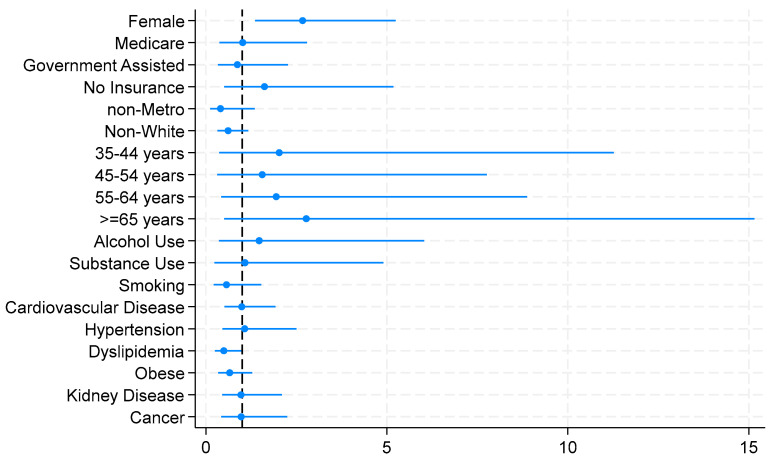
Logistic model for the association between diabetic status and sociodemographics, social habits, and comorbidities.

**Table 1 healthcare-12-00026-t001:** Study sample characteristics.

Variable	Category	Sample N = 206 (100%)	Controlled Diabetes N = 110 (100%)	Uncontrolled Diabetes N = 96 (100%)
Sex	Male Female	105 (51.0%) 101 (49.0%)	47 (42.7%) 63 (57.3%)	58 (60.4%) 38 (39.6%)
Age	<34 years 35–44 years 45–54 years 55–64 years ≥65 years	10 (4.9%) 17 (8.3%) 31 (15.0%) 64 (31.1%) 84 (40.8%)	4 (3.6%) 9 (8.2%) 14 (12.7%) 32 (29.1%) 51 (46.4%)	6 (6.3%) 8 (8.3%) 17 (17.7%) 32 (33.3%) 33 (34.4%)
Race	White Black Hispanic Asian Others	102 (49.5%) 99 (48.0%) 2 (1.0%) 2 (1.0%) 1 (0.5%)	57 (51.8%) 50 (45.5%) 0 (0.0%) 2 (1.8%) 1 (0.9%)	45 (46.9%) 49 (51.0%) 2 (2.1%) 0 (0.0%) 0 (0.0%)
Insurance Type	Private Medicare Government Assisted No Insurance	81 (39.3%) 72 (35.0%) 35 (17.0%) 18 (8.7%)	41 (37.3%) 43 (39.1%) 16 (14.5%) 10 (9.1%)	40 (41.7%) 29 (31.2%) 19 (19.8%) 8 (8.3%)
Area	Metro Nonmetro	191 (92.7%) 15 (7.3%)	105 (95.5%) 5 (4.5%)	86 (89.6%) 10 (10.4%)
Obesity	Normal Overweight Obese	33 (16.0%) 50 (24.3%) 123 (59.7%)	18 (16.4%) 31 (28.2%) 61 (55.5%)	15 (15.6%) 19 (19.8%) 62 64.6%)
Hypertension	No Yes	36 (17.5%) 170 (82.5%)	21 (19.1%) 89 (80.9%)	15 (15.6%) 81 (84.4%)
Cardiovascular	No Yes	131 (63.6%) 75 (36.4%)	73 (66.4%) 37 (33.6%)	58 (60.4%) 38 (50.7%)
Dyslipidemia	No Yes	128 (62.1%) 78 (37.9%)	76 (69.1%) 34 (30.9%)	52 (54.2%) 44 (45.8%)
Kidney Disease	No Yes	167 (81.1%) 39 (18.9%)	90 (81.8%) 20 (18.2%)	77 (80.2%) 19 (19.8%)
Cancer	No Yes	169 (82.0%) 37 (18.0%)	88 (80.0%) 22 (20.2%)	81 (84.4%) 15 (15.6%)
Alcohol Use	No Yes	196 (95.1%) 10 (4.9%)	105 (95.5%) 5 (4.5%)	91 (94.8%) 5 (5.2%)
Substance Use	No Yes	197 (95.6%) 9 (4.4%)	106 (96.4%) 4 (3.6%)	91 (94.8%) 5 (5.2%)
Smoking	No Yes	180 (87.4%) 26 (12.6%)	100 (90.9%) 10 (9.1%)	80 (83.3%) 16 (16.7%)
Total Comorbidity	0 1 2 3 4 5 6	10 (4.9%) 28 (13.6%) 29 (28.6%) 68 (33.0%) 34 (16.5%) 6 (2.9%) 1 (0.5%)	5 (5.2%) 10 (10.4%) 24 (25.0%) 34 (35.4%) 17 (17.7%) 5 (5.2%) 1 (1%)	5 (4.5%) 18 (16.4%) 35 (31.8%) 34 (30.9%) 17 (15.5%) 1 (0.9%) 0 (0.0%)
Concordant Comorbidity	0 1 2 3 4	14 (6.8%) 41 (19.9%) 68 (33.0%) 63 (30.6%) 20 (9.7%)	9 (8.2%) 25 (22.7%) 39 (35.5%) 30 (27.3%) 7 (6.4%)	5 (5.5%) 16 (16.7%) 29 (3.2%) 33 34.4%) 13 (13.5%)
Discordant Comorbidity	0 1 2	139 (67.5%) 58 (28.2%) 9 (4.4%)	73 (66.4%) 32 (29.1%) 5 (4.5%)	66 (68.8%) 26 (27.1%) 4 (4.2%)

**Table 2 healthcare-12-00026-t002:** Logistic regression results for the relationship between diabetic status and sociodemographic characteristics, social factors, and comorbidity.

Variable	Categories	AOR	*p*-Value	95% C.I.	
				Lower	Upper
Sex	Male	--	--	--	--
	Female	2.673	0.004 *	1.363	5.239
Insurance Type	Private	--	--	--	--
	Medicare	1.017	0.974	0.371	2.789
	Government Assisted	0.870	0.776	0.335	2.262
	No Insurance	1.617	0.418	0.505	5.179
Location	Metro	--	--	--	--
	Non-metro	0.404	0.141	0.121	1.349
Race	White	--	--	--	--
	Non-white	0.613	0.138	0.322	1.170
Age Groups	<34 years	--	--	--	--
	35–44 years	2.027	0.420	0.364	11.271
	45–54 years	1.556	0.590	0.312	7.763
	55–64 years	1.942	0.392	0.424	8.874
	>=65 years	2.773	0.239	0.507	15.156
Alcohol Use	No	--	--	--	--
	Yes	1.472	0.591	0.359	6.028
Substance Use	No	--	--	--	--
	Yes	1.081	0.920	0.238	4.901
Smoking	No	--	--	--	--
	Yes	0.570	0.263	0.213	1.524
Cardiology	No	--	--	--	--
	Yes	0.989	0.976	0.510	1.921
Hypertension	No	--	--	--	--
	Yes	1.071	0.873	0.459	2.499
Dyslipidemia	No	--	--	--	--
	Yes	0.495	0.041 *	0.252	0.972
Obese	No	--	--	--	--
	Yes	0.657	0.216	0.338	1.278
Renal Disease	No	--	--	--	--
	Yes	0.968	0.935	0.446	2.100
Cancer	No	--	--	--	--
	Yes	0.975	0.952	0.423	2.244

* Indicating statistical significance at 0.05.

**Table 3 healthcare-12-00026-t003:** Relationship of diabetic status with concordant and discordant comorbidity, results from ordered logistic regression.

Variable	AOR	*p*-Value	95% C.I.	
			Lower	Upper
Concordant comorbidity (0–4)	0.741	0.027 *	0.569	0.967
Discordant comorbidity (0–2)	1.050	0.845	0.642	1.719

* Indicating statistical significance at 0.05.

## Data Availability

The data supporting this study’s findings are available from the Augusta University Human Research Box Folder. Restrictions apply to the availability of these data, which were used upon IRB approval for this study and thus are not publicly available.

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
