# Peer review of "Factors Associated with Controlled Glycemic Levels in Type 2 Diabetes Patients: Study from a Large Medical Center and Its Satellite Clinics in Southeast Region in the USA"

_healthcare, 2023, doi:10.3390/healthcare12010026_

Round 1

Reviewer 1 Report

Comments and Suggestions for Authors

Major comments:

Whether duration of diabetes show significant differences between controlled diabetes and uncontrolled diabetes.

Page 3 Line 120 to 122, please refer to the specific reference or guideline support utilizing A1C 7% as cutoff for distiguishing controlled from uncontrolled.

Page 3 Line 131. Please make sure the rationel for choosing BMI >29 as obese definition, instead of 30.

Table 1. Please carefully double check the data across the whole table. eg. In the row of total comorbidity, the sum of percentage of all the seven groups (0-6) is not equal to 100% for uncontrolled diabetes.

Minor comments:

Table 1. Please confirm the use of N, n or m to indicate sample size in each group.

Table 2. Make sure the expression "Race (2)" is correct.

Author Response

Thank you for your time to review our manuscript and your comments in the effort to improve our paper. Please see our responses attached.

Reviewer 2 Report

Comments and Suggestions for Authors

The manuscript “Factors Associated with Controlled Glycemic Levels in Type 2 Diabetes Patients: Study from a Large Medical Center and Its Satellite Clinics in Southeast Region in the USA” describes a retrospective cohort study with the goal of determining which sociodemographic and comorbid factors may be associated with glycemic control. The researchers assembled a cross-section of 206 patients with type 2 diabetes in the year 2019 from a primary dataset of medical records from a health system in the Southeast United States. The primary specification strategy is a multivariable logistic regression in which the dependent variable is whether or not a patient’s diabetes is controlled. Overall, the researchers employed appropriate methods to address the research question, and the conclusions are consistent with the results of the analysis. While the research question is certainly relevant to the target audience of Healthcare, the manuscript in its current form does not necessarily address a gap in the literature, but its findings are consistent with an existing body of evidence regarding glycemic control. There are a few areas, described below, where the researchers could potentially expand on and improve the analysis to better differentiate this manuscript from the existing literature.

Be cautious with the use of words like “determinants” of diabetic status (lines 14, 20, 82, 196, 211, 215, 224, 263). Given the descriptive (non-causal) nature of this study, you cannot definitively say what causes controlled or uncontrolled diabetes. You do not specify any control variables for potential confounders, so it’s possible these relationships are driven by some other factor. When describing these relationships you should consider re-phrasing the language to something like “factors associated with controlled status” or “factors that make patients more likely to have controlled diabetes.”

Along those same lines, I would remove that statement in lines 17-18 altogether (“A typical controlled diabetic…”). This is a confusing statement—it is not borne out by the results in Table 2. Instead consider just listing the two factors that had meaningful associations (i.e. sex and dyslipidemia were found to be correlated with diabetic status). 

The statement in lines 71-73 (“Despite major public health impact… little information on diabetic care outcomes exist.”) is at odds with a lot of the research cited in your discussion section. There is a significant body of literature describing predictors of diabetic outcomes. Where your study may be novel, is in the use of your specific dataset-- a single organization medical records review. So I would recommend trying to distinguish your study in that way. In other words, what might this study say about diabetics living in the Augusta, GA area? And consider how the population of this organization may be similar or different to the broader U.S. population.

Regarding your study sample, if you have access to the full 4,000+ patients with T2D you could do some interesting subgroup analyses that would add to the literature. Perhaps the relationship between sex and controlled status varies by racial/ethnic group or by age group. You also mention that they patients visited AUH for various reasons such as primary/follow-up care, specialty care, post-op, etc. If you have access to that data you could also potentially use those as independent variables in your analysis (or future analyses). This could meaningfully add to the literature to answer questions like “are T2D patients visiting the emergency room more likely to be uncontrolled?”

Double check your power calculations. The population input should be the population of available data that you are drawing a sample from. In this case it would be the 4,071 T2D patients seen at AUH.

Line 140 should read “independent variables.”

It’s not immediately obvious if all of the variables in Table 2 were included in a single regression model or if each of these relationships was tested separately.

There are some relationships between sociodemographics and diabetic control that are established in the literature that were not demonstrated here. For instance, the link between insurance status and glycemic control is well-documented in larger population-level studies. It makes sense that patients who have insurance coverage might have better access, feel more empowered to manage their care, have consistent, affordable access to medication, etc. That relationship may not have been picked up here because there may be something unique about Augusta’s population or AUH’s payer mix—and you do point to that in your discussion regarding the study’s limited generalizability. However, you might consider specifying insurance status as a control variable rather than an independent variable. This would indicate to the reader that you’re accounting for potential confounder. The outcome of your regression would be the same, you just wouldn’t interpret the coefficients on insurance status, nor would you report them in Table 2.

Author Response

(The authors gave the same response as above.)

Reviewer 3 Report

Comments and Suggestions for Authors

In this work, the authors investigate factors associated with type 2 diabetes according to the glycemic control, expressed by HbA1c. On the whole, this retrospective study is well designed and conducted.

My major observations are the following:

- Lines 120-122: Please specify which technique/method has been used to assess HbA1c  and its analytical performance in terms of precision.

- Lines 216-217: Gender differences in terms of HbA1c are debated (Ahmed SF., Life 2023) and there are reports in diabetic patients stating the opposite of what described by the authors. (Vaccaro O, et al, Atherosclerosis 2008; Duarte FG, et al, BMJ Open 2018  ). Differences can be attributed to different populations and due to socio-cultural factors Please discuss.

- In the discussion, it could be of interest mentioning how glycemic control  may influence the status of inflammation and thrombophilia in type 2 diabetes (Palella E et al, Int J Environ Res Public Health, 2020), both conditions predisposing to cardiovascular risk. 

- Bibliography could be improved and expanded. References 1 and 2 could be substituted or flanked by other, more representative ones. 

Minor point:

Line 77: please remove “( Kazemian et al., 2019)”, which already corresponds to reference 25.

Author Response

(The authors gave the same response as above.)
